# Relative Effect of Extracorporeal Shockwave Therapy Alone or in Combination with Noninjective Treatments on Pain and Physical Function in Knee Osteoarthritis: A Network Meta-Analysis of Randomized Controlled Trials

**DOI:** 10.3390/biomedicines10020306

**Published:** 2022-01-28

**Authors:** Chun-De Liao, Yu-Yun Huang, Hung-Chou Chen, Tsan-Hon Liou, Che-Li Lin, Shih-Wei Huang

**Affiliations:** 1Master Program in Long-Term Care, College of Nursing, Taipei Medical University, Taipei 110301, Taiwan; 08415@s.tmu.edu.tw; 2Department of Physical Medicine and Rehabilitation, Shuang Ho Hospital, Taipei Medical University, New Taipei City 235041, Taiwan; 10462@s.tmu.edu.tw (H.-C.C.); peter_liou@s.tmu.edu.tw (T.-H.L.); 3Department of Pediatrics, New York University Langone Medical Center, New York, NY 10016, USA; huang-yu-yun@hotmail.com; 4Department of Physical Medicine and Rehabilitation, School of Medicine, College of Medicine, Taipei Medical University, Taipei 110301, Taiwan; 5Department of Orthopedic Surgery, Shuang Ho Hospital, Taipei Medical University, New Taipei City 23561, Taiwan; 11010@s.tmu.edu.tw; 6Department of Orthopedics, School of Medicine, College of Medicine, Taipei Medical University, Taipei 11031, Taiwan

**Keywords:** osteoarthritis, shockwave therapy, pain, function, inflammation

## Abstract

Extracorporeal shockwave therapy (ESWT) has been recommended for managing pain in patients with knee osteoarthritis (KOA). The difference in therapeutic effects between radial shockwave characteristics (RaSW) and focused shockwave characteristics (FoSW) with different energy levels for KOA remains controversial. The purpose of this network meta-analysis (NMA) was to identify the effects relative to the different ESWT regime and combination treatments on pain and functional outcomes in individuals with KOA. The randomized controlled trials (RCTs) which investigated the efficacy of RaSW, FoSW, and combination treatments in patients with KOA were identified by searches of electronic databases. The included RCTs were analyzed through NMA and risk-of-bias assessment. We analyzed 69 RCTs with a total of 21 treatment arms in the NMA. Medium-energy FoSW plus physical therapy, medium-energy acupoint RaSW plus Chinese medicine, and high-energy FoSW alone were the most effective treatments for reducing pain [standard mean difference (SMD) = −4.51], restoring function (SMD = 4.97), and decreasing joint inflammation (SMD = −5.01). Population area and study quality influenced the treatment outcomes, particularly pain. Our findings indicate that medium-energy ESWT combined with physical therapy or Chinese medicine is beneficial for treating pain and increasing function in adults with KOA.

## 1. Introduction

Knee osteoarthritis (KOA) is a serious joint disease and prevalent chronic musculoskeletal disorder, with pain being its primary symptom and main clinical presentation [1,2]. In KOA, knee pain occurs at an early stage of the disease and is gradually aggravated throughout disease progression [3]. With disease progression, KOA impairs musculoskeletal system [4], ultimately leading to physical difficulty [2,5,6]. Such musculoskeletal pain in KOA is closely associated with a decline in health state and a negatively impact on quality of life [7,8,9]. In addition, the most common practical problem of KOA is the pain-induced limitation in physical function and mobility, especially walking ability [10,11,12], stair negotiation [13], and postural transition in activities [10,11,12]. Under such circumstances, the development of efficient treatment strategies for pain management is essential for individuals with KOA.

Extracorporeal shockwave therapy (ESWT) is a convenient, cost-effective treatment for managing pain in common musculoskeletal conditions of the lower limbs [14,15,16,17,18]. ESWT is safely used in clinical practice and serves as an alternative to conservative injections or surgery owing to its noninvasive and effective application in musculoskeletal disorders [19,20]. In addition, ESWT could play a role in regenerative medicine through its stimulation of soft tissue healing [21,22,23] and cartilage regeneration [24,25] and inhibition of pain receptors [26]. On the basis of the sources of energy production and the delivery pathway that propagates acoustic energy through biological tissue, ESWT can be characterized into two types, namely focused shockwave (FoSW) and radial shockwave (RaSW) [27,28,29]. FoSW and RaSW use different energy sources to generative shockwave impulses [29,30,31,32,33], and each type of two ESWT applications should be served as an independent treatment modality [29,31,32]. The effects of ESWT depend on the energy level which is presented as energy flux density (EFD, mJ/mm^2^) per shockwave impulse [28,30]. The energy level of ESWT ranges from 0.001 to 0.5 mJ/mm^2^ [30,31,34,35]. Identifying relative effects among different shockwave applications is important due to that an overly high dosage may have high risks of poor treatment outcome and adverse events [36].

A number of systemic review and meta-analysis studies have investigated the efficacy of ESWT in the treatment of KOA [37,38,39,40,41,42,43]. Most trials included in these reviews and meta-analysis studies conducted an ESWT intervention in combination with noninjection treatments such as conventional physical therapy (CPT) and traditional Chinese medicine (TCM). However, few of the systemic reviews compared ESWT with other interventions [40,41,42,43]. The relative effects of the various combination treatment regimens composed of different ESWT applications (i.e., FoSW versus RaSW and low versus high EFD) and noninjective treatments, with reference to usual care (UC), have yet to be determined. In addition, all of these systemic reviews investigated the effects of ESWT on perceived pain and function outcomes [37,38,39,40,41,42,43], among which only two reported the combined effects on joint range of motion and walking performance [41,43]; none focused on the disease inflammation outcomes of arthritic knees. Therefore, in this study, the effects of ESWT on physical mobility, joint function, and disease inflammation were investigated in individuals with KOA.

The relative effects among different combination treatment regimens of various ESWT applications remain unclear. Therefore, the purpose of this study was (1) to identify the relative effects of different ESWT applications and combination treatment regimens on pain outcome, global function, and disease inflammation through network meta-analysis (NMA) and (2) to determine the optimal treatment strategy by using the ranking probabilities of each intervention type for individuals with KOA.

## 2. Materials and Methods

### 2.1. Study Design

The present NMA study was conducted in accordance with the guidelines of the Preferred Reporting Items for Systematic Reviews and Meta-Analyses Statement (PRISMA) [44] and the additional statement for NMA [45,46]. A comprehensive electronic search of online sources was performed to identify eligible randomized controlled trials (RCTs) reporting the efficacy of ESWT for KOA. All of the articles were identified from electronic databases, including the Physiotherapy Evidence Database (PEDro), PubMed, Cochrane Library Database, Embase, China Knowledge Resource Integrated Database, and Google Scholar. In addition, secondary sources included trials enrolled in previous systemic reviews retrieved from the aforementioned sources. There was no limitation of the language or publication year, which minimized language and publication bias. At the beginning of search, all potentially eligible articles were independently searched by two team members, CDL and SWH, who followed the criteria of study section to screen relevant articles. A consensus meet was performed to resolve any disagreement between two authors. The protocol of this NMA was registered at PROSPERO (registration number: CRD42021292060).

### 2.2. Search Strategy

The following keywords relating to individuals’ conditions were used: “osteoarthritis” OR “gonarthritis” OR “degenerative cartilage.” The following keywords relating to interventions were used: “extracorporeal shock wave therapy” OR “shockwave therapy.” The detailed search formulas for each database are presented in online Appendix A.

### 2.3. Selection Criteria of Studies

Articles which met the following criteria were included (Table 1): (1) the trial was conducted based on a design of two-arm or multiarm RCT, as well as a design of quasi RCT; (2) the trial recruited such patients who had a symptom or radiographic diagnosis of KOA; (3) treatment groups received ESWT alone or in combination with other noninjective treatments such as CPT and TCM; (4) the control group received a placebo ESWT, relatively low-dosage ESWT, or non-ESWT comparator intervention. Non-ESWT comparator interventions included CPT, TCM, and pain medication, all of which were classified as UC in this study; and (5) the study reported at least one of the primary or secondary outcome measures defined in Section 2.4.

Studies were eliminated if (1) the trial (or study arm) used intra-articular injections as a primary or comparator intervention; (2) the trial was conducted in vitro or in vivo by using an animal model; or (2) they were non-RCTs including case reports, case series, or prospectively designed trials without a comparison group.

### 2.4. Outcome Measures

The primary outcomes in this NMA included measures of pain and patient-perceived global function. Pain score was assessed using a quantifiable scale such as a visual analogue scale [44] or pain subscale derived from questionnaire-based instruments [45]. Global function was measured using self-administered instruments [45], which included the Western Ontario and McMaster Universities Arthritis Index (WOMAC) [46], Knee Injury and Osteoarthritis Outcome Score [47], Lequesne index [48], and Lysholm Knee Scoring Scale [49]. When the WOMAC total score was unavailable, the total sum of the scores derived from its subscales (i.e., pain, stiffness, and physical difficulty) were calculated, and other assessment tools were employed to measure global function. If the trial reported two or more global function scores, the priority of selection for analyses was as follows: WOMAC, Knee Injury and Osteoarthritis Outcome Score, Lysholm Knee Scoring Scale, and Lequesne index [50,51].

The secondary outcome was the disease inflammation of KOA. Disease inflammation was measured using the intra-articular level of the inflammatory factors as follows: (1) proinflammatory cytokines such as interleukin 1 beta, tumor necrosis factor α, and interleukin 6, which are involved in the pathogenesis of KOA [52]; (2) nitric oxide, which is associated with the disease progression of KOA [53,54]; and (3) synovial fluid adipokines such as chemerin, which are associated with the disease severity of KOA [55,56].

### 2.5. Data Extraction and Synthesis

The following data were extracted from each included trial and presented in an evidence table (Table 2): (1) characteristics of study design and samples including the study arm, age, body mass index (BMI), sex distribution, and geographic area of the study population; (2) characteristics of disease onset including affected side, disease severity (Kellgren and Lawrence grade), and disease duration; (3) measured time points; and (4) main outcomes. If the trial had multiple treatment or control groups, only the study arms not related to intra-articular injections were selected and extracted for analyses. One of the team members, CDL, recorded the relevant data derived from the included RCTs, and the second team member, HCC, reviewed and confirmed the extracted data. If there was any discrepancy between the two team members, it was discussed and resolved through a consensus meeting. If the disagreements were not resolved, a third team member, CLL, was consulted for further judgement.

If the RCT separately reported treatment effects on bilateral legs, those of bilateral legs were combined to enable a single comparison, and such a procedure is recommended in the Cochrane Handbook for Systematic Reviews of Interventions [57]. The energy level of the ESWT was defined based on the EFD administered in each included trial and was classified as low (<0.08 mJ/mm^2^), medium (≥0.08 mJ/mm^2^, <0.25 mJ/mm^2^), or high (≥0.25 mJ/mm^2^) [58,59,60,61]. If the trial had multiple study arms with different EFDs, results of the same energy level were combined into a single treatment effect [57]. The duration of follow-up was assessed and defined as immediate (<1 month), short (≥1 and <3 months), medium (≥3 and <6 months), or long term (≥6 months) for subgroup analysis; when multiple time points were reported within the same timeframe, the analyzed results constituted those with the longest follow-up period for each of the included studies. For example, if the measured time points for pain score were 12 and 16 weeks in one trial, only the data from the 16-week follow-up were selected as medium-term results.

The presentation of the all-cause withdrawal rate was expressed to assess tolerance to the ESWT regimen. We also examined adverse events when reported; however, they were not specified a priori.

### 2.6. Methodological Quality and Risks of Bias of Included Trials

In this NMA, the PEDro quality score was used to rank the methodological quality and risk of bias. Two of the team members, CDL and SWH, independently assessed methodological quality of each included RCT. The PEDro scale comprises 10 ranking items which correspond to selection bias (items 1–3: random allocation, concealed allocation, similarity at baseline); performance bias (items 4–5: subject blinding, therapist blinding); detection bias (items 6 and 10: assessor blinding, point and variability measures for at least one key outcome); attrition bias (items 7–8: >85% follow-up for at least one key outcome, intention-to-treat analysis); and reporting bias (item 9: between-group statistical comparison for at least one key outcome). Validity of the PEDro scale has been identified [62]. The interrater reliability of ratings for the individual PEDro scale items varies from moderate to excellent (Kappa value: 0.53–0.94) for assessing the quality of RCTs [63]. In addition, an intraclass correlation coefficient for the PEDro total sum score has been identified as 0.91 [95% confidence interval (CI): 0.84–0.95] [64]. The methodological quality of the included RCTs was considered as low, medium, and high with a total PEDro score ≤3/10, 4–6/10, and ≥7/10, respectively [65].

A funnel plot was used to identify potential publication bias by subjective visual assessment [66]. In addition, Egger’s regression test was performed to identify any asymmetry in the funnel plot [67], which explored reporting bias.

### 2.7. Statistical Analysis

The effect sizes on each outcome measure between any two study arms were computed in this NMA. The effect size was expressed as standard mean difference (SMD), which was calculated by dividing the between-group mean difference in the change score by the pooled standard deviation (SD). To partially correct between-participant variability, all analyses were performed based on change scores (i.e., change from baseline) [57]. Where the change score in mean and SD was reported it was directly collected from the included RCT. If SD of change score was not reported for the outcome measure, it was estimated by the baseline and posttest measured SD in accordance with the Cochrane Handbook for Systematic Reviews of Interventions [57]. We followed Rosenthal’s approach by assuming a within-participant correlation coefficient of 0.7 between the baseline and posttest measured data [68]. Where *p* values or 95% CIs were reported instead of SDs, from which the SDs were calculated using the methods recommended by the Cochrane Handbook for Systematic Reviews of Interventions [57]. For trials reporting data as the median and interquartile range, the median was used to be representative of the mean value, and the interquartile range was divided by 1.35 to produce the SD [57].

The random-effects NMA were performed for all outcome measures using the frequentist methods. All analyses were performed using statistical software R (version 4.0.4, R Foundation for Statistical Computing, Vienna, Austria) [69,70]. Direct and indirect comparisons between different ESWT regimens were performed [71]. The Cochran’s Q statistics or *I*^2^ test was employed to assess heterogeneity along with *τ*^2^ values to estimate the variance across studies. In addition, we assessed inconsistency between direct and indirect comparisons using the node-splitting method [72,73]. Statistical significance was set at a two-way *p* value less than 0.05.

P score was used to rank the probabilities of effect estimation for each treatment [74]. Network forest plots which presented relative effects among treatment options using UC as reference were generated to identify the uncertainty in NMA [75].

Network metaregression analyses were performed to assess the confounding effects of potential moderators based on (1) participant characteristics including age, BMI, sex distribution (i.e., proportion of female participants in a sample), disease-onset duration, and area of the study population; (2) study methodology including level of methodological quality (i.e., PEDro score) and follow-up duration; and (3) intervention design including shockwave type, EFD level, treatment composition (i.e., monotherapy or combination treatment), and treatment duration.

Potential publication bias was assessed through the visual inspection of a funnel plot [76], and Egger’s regression asymmetry test was performed to explore possible reporting bias [67].

## 3. Results

### 3.1. Patient Demographics and Clinical Characteristics

Figure 1 presents the selection process of eligible trials. A total of 800 articles were identified through an electronic and manual literature search, after which the duplicates were excluded. By reviewing the titles and abstracts, 238 studies were assessed for their eligibility, among which 114 were considered relevant for full-text assessment. Finally, the sample in this NMA comprised 70 articles on 69 RCTs published between 2000 and 2020.

### 3.2. Study Characteristics

The study characteristics of and patient demographic data from the included RCTs were summarized in Table 2; the details of each trial are presented in Appendix A. A total of 5980 participants who had received a diagnosis of symptomatic or radiographic KOA were recruited. Overall, the whole sample had a mean age of 59.8 (range: 40.1–80.3) years, mean BMI of 25.7 (range: 22.3–36.4) kg/m^2^, and mean disease duration of 52.4 (range: 6–200) months; the average proportion of female participants was 60.5% (range: 10–94.4%), which was estimated through the exclusion of eight sex-specific (female participants only) RCTs.

In this NMA, 54 of the included RCTs were two-arm studies, and the other 15 RCTs had a multiarm design, with a total of 155 study arms (ESWT, 94 arms). Among all participants, 2234 (37.4%) received diet therapy alone, 1502 (25.1%) received combination treatment, and 2244 (37.5%) received UC (i.e., placebo or non-ESWT comparator). With respective to the follow-up duration, 67 RCTs had an immediate or short-term follow-up duration of <12 weeks, 21 had a medium-term follow-up duration ranging from 12 to 19 weeks, and 10 had a long-term follow-up duration ranging from 6 to 14 months (Appendix A).

### 3.3. ESWT Intervention Characteristics

#### 3.3.1. ESWT Protocols

The protocols for ESWT intervention employed in the included RCTs are summarized in Appendix A. With respect to shockwave types, 14 (20.2%) and 48 (69.5%) out of the 69 included RCTs employed FoSW and RaSW, respectively, targeting tender points around the knee joint. Specifically, seven RCTs (10.3%) applied acupoint therapy by using RaSW for patients with KOA. Regarding the shock energy level, 31 RCTs applied ESWT with an EFD of ≥0.25 mJ/mm^2^, 32 used medium EFD, and nine employed low EFD (Appendix A). In total, 65 RCTs applied an ESWT protocol consisting of three to 14 treatment sessions (one to three sessions weekly) within a treatment duration of 2 to 8 weeks, whereas 4 RCTs administered 15 to 30 treatment sessions within a treatment duration of 9 to 24 weeks. No local anesthesia was administered at the treatment site during application in all the included RCTs.

#### 3.3.2. Treatment Arms of ESWT

In summary (Table 3), a total of 21 treatment arms of ESWT were identified and included in the NMA, which was based on three types of ESWT therapy (FoSW, RaSW, and acupoint ESWT), three levels of EFD (high, medium, and low), and three combination treatment regimens (ESWT alone, ESWT plus CPT, and ESWT plus TCM).

### 3.4. Risk of Bias in Included Studies

The detailed ratings of PEDro items of each included RCT are presented in Appendix A. Overall, results of methodological quality assessment showed that 25 out of the 69 (36.2%) included RCTs were classified as high methodological quality whereas the other 44 RCTs were considered as medium. The median PEDro score was estimated as 6 out of 10 (range: 5/10 to 9/10) with an intraclass correlation coefficient of 0.98 (95% CI: 0.96–0.98). In the view of the risk of bias across RCTs, all the 69 included RCTs employed random allocation, similarity at baseline, between-group comparisons, and point estimates and variability. In total, 16 (64%), 2 (8%), and 23 (92%) out of the 25 high-quality RCTs blinded the participants, therapists, and assessors, respectively, whereas only one medium-quality RCTs performed assessor blinding; none of the medium-quality RCTs blinded the participants or therapists. Moreover, 12 of the 25 (48%) high-quality RCTs performed allocation concealment, as did one medium-quality RCT.

### 3.5. Effectiveness of Treatment for Pain Reduction Assessed in NMA

Figure 2 shows the network of eligible comparisons between any two of the treatment options for each outcome measure. Figure 3 and Figure 4 present the relative effects of ESWT treatment on primary outcomes during an overall follow-up duration and at each time frame, and Appendix A present the details of each comparison, respectively. In addition, Appendix A demonstrate the league tables which provide results of pairwise meta-analysis and NMA.

#### 3.5.1. Pairwise Meta-Analysis

Direct comparisons of pairwise meta-analyses (Appendix A) indicated that FoSW applied with medium (SMD = −1.98; 95% CI: −3.03, −0.92) and high (SMD = −1.21; 95% CI: −2.15, −0.26) EFD were more efficacious than UC for pain reduction; in addition, medium-EFD (SMD = −1.64; 95% CI: −2.60, −0.69) and high-EFD (SMD = −1.68; 95% CI: −2.29, −1.07) RaSW induced greater pain-related changes compared with UC and the combination treatments [medium-EFD RaSW plus CPT (SMD = −1.01; 95% CI: −1.88, −0.15); high-EFD RaSW plus either CPT (SMD = −2.20; 95% CI: −3.30, −1.09) or TCM (SMD = −1.28; 95% CI: −2.47, −0.09)].

In addition, the combination treatments yielded greater reductions in pain compared with ESWT alone; such results were observed in medium-EFD FoSW plus CPT (SMD = −2.56; 95% CI: −4.67, −0.45) and high-EFD RaSW plus TCM (SMD = −1.71; 95% CI: −2.91, −0.51) in comparison to medium-EFD FoSW and high-EFD RaSW alone, respectively.

#### 3.5.2. Global Effects in NMA

The NMA for pain score was based on 62 RCTs (55 two-arm RCTs; 7 three-arm RCTs) with 22 treatments, 32 designs, and 76 pairwise comparisons. The NMA results demonstrated that, in comparison with UC, FoSW with medium (SMD = −1.95; 95% CI: −2.95, −0.95) and high (SMD = −1.21; 95% CI: −2.15, −0.26) EFD induced greater changes in pain score—as did the RaSW with medium (SMD = −1.61; 95% CI: −2.36, −0.86) and high (SMD = −1.64; 95% CI: −2.19, −1.10) EFD—during the overall follow-up duration; additionally, acupoint ESWT using medium-EFD RaSW (SMD = −1.64; 95% CI: −2.19, −1.10) resulted in greater reductions in pain score compared with UC (Figure 3 and Appendix A). The combined effects of medium-EFD FoSW plus CPT (SMD = −4.51), high-EFD RaSW plus either CPT (SMD = −1.74) or TCM (SMD = −2.29), and acupoint ESWT plus TCM (SMD = −3.13) on pain reduction relative to UC were stronger than the effect of the solely FoSW, RaSW, and acupoint ESWT, respectively (Figure 3 and Appendix A).

During the overall follow-up duration, medium-EFD FoSW plus CPT was ranked the most effective (P score = 0.97) among all treatment arms for pain reduction—followed by acupoint ESWT plus TCM (P score = 0.84), high-EFD RaSW plus TCM (P score = 0.74), and low-EFD FoSW plus CPT (P score = 0.66; Appendix A). A significant global heterogeneity was observed (*τ*^2^ = 1.07, *I*^2^ = 94.2%, *p* < 0.0001). According to the results of node-splitting analyses (Appendix A), there were no inconsistencies between the direct and indirect evidence for each comparison.

#### 3.5.3. Subgroup Analysis of Follow-Up Duration

The combination treatment of medium-EFD FoSW plus CPT (SMD = −4.02; P score = 0.95) and medium-EFD FoSW alone (SMD = −2.43; P score = 0.88) were ranked as optimal treatments for pain reduction at the immediate and medium-term follow-up duration, respectively, whereas acupoint ESWT (i.e., medium-EFD RaSW) plus TCM (SMD = −2.83; P score = 0.83) and medium-EFD RaSW (SMD = −2.40; P score = 0.96) were ranked the highest among all treatments over the short-term and long-term follow-up durations, respectively (Figure 3 and Appendix A).

In summary, we determined that the combination treatments were the optimal ESWT protocols at immediate and short-term follow-up, whereas medium-EFD ESWT exhibited a higher ranked treatment effect on pain reduction at each follow-up timeframe, irrespective of the type of ESWT and its combination non-ESWT treatments.

### 3.6. Effectiveness of Treatment for Global Function

#### 3.6.1. Pairwise Meta-Analysis

Direct comparisons of pairwise meta-analyses (Appendix A) indicated that FoSW applied with medium (SMD = 1.56, 95% CI: 0.23–2.88) and high (SMD = 1.41, 95% CI: 0.38–2.45) EFD were more efficacious than UC for function recovery, as were medium-EFD (SMD = 2.79, 95% CI: 1.72–3.86) and high-EFD (SMD = 1.65, 95% CI: 1.01–2.29) RaSW; similar results were observed in the combined treatment medium-EFD RaSW plus CPT (SMD = 1.61, 95% CI: 0.73–2.49), as well as high-EFD RaSW combined with either CPT (SMD = 1.61, 95% CI: 0.43–2.80) or TCM (SMD = 1.88, 95% CI: 0.57–3.19).

In addition, acupoint ESWT plus TCM induced greater changes in function restoration than acupoint ESWT alone (SMD = 2.33, 95% CI: 0.03–4.63); similar results were observed in the combination treatments of high-EFD RaSW plus CPT (SMD = 3.32, 95% CI: 0.96–5.69) and high-EFD RaSW plus TCM (SMD = 1.44, 95% CI: 0.12–2.75) compared with high-EFD RaSW alone (Appendix A).

#### 3.6.2. Global Effects of NMA

The NMA for global function was based on 63 RCTs (55 two-arm RCTs; eight three-arm RCTs) with 21 treatments, 32 designs, and 79 pairwise comparisons. The NMA results indicated that, during the overall follow-up duration, FoSW employed with medium (SMD = 1.44; 95% CI: 0.24–2.64) and high (SMD = 1.41; 95% CI: 0.38–2.45) EFD exerted significant effects on function regain relative to UC, as did the medium-EFD (SMD = 2.11; 95% CI: 1.28–2.93) and high-EFD (SMD = 1.54; 95% CI: 0.96–2.12) RaSW (Figure 4 and Appendix A); additionally, acupoint ESWT with medium-EFD RaSW led to a greater increase in function scores compared with UC (SMD = 2.46; 95% CI: 1.21–3.71). The combined effects of medium-EFD FoSW plus CPT (SMD = 2.29), high-EFD FoSW plus CPT (SMD = 1.97), high-EFD RaSW plus TCM (SMD = 2.50), and acupoint ESWT plus TCM (SMD = 4.97) on function restoration relative to UC were stronger than the effect of FoSW alone, RaSW alone, and acupoint ESWT alone, respectively (Figure 4 and Appendix A).

Among all treatment arms for global function, acupoint ESWT plus TCM was ranked the most effective (P score = 0.97) followed by acupoint ESWT plus CPT (SMD = 3.95; P score = 0.87), high-EFD RaSW plus TCM (P score = 0.78), and acupoint ESWT alone (SMD = 2.46, P score = 0.77; Figure 4 and Appendix A). A significant global heterogeneity was observed (*τ*^2^ = 1.30, *I*^2^ = 95%, *p* < 0.0001). According to the results of node-splitting analyses (Appendix A), there were no inconsistencies between the direct and indirect evidence for each comparison.

#### 3.6.3. Subgroup Analysis of Follow-Up Duration

Acupoint ESWT in combination with TCM was ranked the highest in terms of immediate (SMD = 2.78; P score = 0.90) and short-term (SMD = 4.17; P score = 0.96) treatment efficacy, respectively, for function restoration among all treatments (Figure 4 and Appendix A). Additionally, medium-EFD RaSW plus CPT resulted in the highest medium-term treatment efficacy for global function (SMD = 5.75, 95% CI: 2.57−8.92; P score = 0.99), as did medium-EFD RaSW alone in long-term follow-up (SMD = 3.70, 95% CI: 2.18−5.21; P score = 0.93).

### 3.7. Effectiveness of Treatment for Disease Inflammation

#### 3.7.1. Pairwise Meta-Analysis

Direct comparisons of the pairwise meta-analyses (Appendix A) indicated that FoSW applying high EFD (SMD = −5.01; 95% CI: −6.98, −3.03) was more efficacious than UC for reducing disease inflammation.

#### 3.7.2. Global Effects of NMA

The NMA for disease inflammation was based on 10 RCTs with two study arms and one RCT with a three-arm design. The NMA results demonstrated that, during the overall follow-up duration, high-EFD FoSW (SMD = −5.01; 95% CI: −6.98, −3.03) exerted significant effects in terms of inflammation reduction relative to UC (Figure 5). High-EFD FoSW was ranked the most effective (P score = 0.98) among all treatment arms for disease inflammation followed by acupoint ESWT plus TCM (SMD = −2.21; P score = 0.68) and medium-EFD RaSW plus TCM (SMD = −1.39; P score = 0.53; Figure 5). A significant global heterogeneity was observed (*τ*^2^ = 1.59, *I*^2^ = 97.2%, *p* < 0.0001). According to the results of node-splitting analyses (Appendix A), there were no inconsistencies between the direct and indirect evidence for each comparison.

### 3.8. Network Metaregression Results for Moderators of Treatment Efficacy

The NMA results are presented in Appendix A. We observed that area of population (*β* = 0.87; *p* < 0.05) and PEDro score (*β* = −0.88; *p* < 0.05) influenced ESWT efficacy for pain reduction. No moderator influenced treatment efficacy in terms of global function and disease inflammation.

### 3.9. Compliance and Side Effects

The rates of compliance with the ESWT interventions were 98%, 95%, and 100% among the included RCTs that reported adherence to ESWT alone, ESWT plus CPT, and ESWT plus TCM, respectively, regardless of shockwave type or EFD level (Table 2).

No serious adverse events or severe complications were observed after ESWT alone or its combination treatments in all of the included RCTs. In total, 11 of the 69 (15.9%) included RCTs reported side effects related to ESWT interventions, the most common of which related to treatment-induced knee pain, swelling, transient subcutaneous congestion, and short-term skin irritation (Appendix A).

### 3.10. Publication Bias

The risk of publication bias across the included RCTs was considered low, since the distribution of the main outcomes in funnel plots did not show asymmetries (Appendix A). Egger’s test results for pain outcome (*p* < 0.0001; Appendix A) and global function (*p* = 0.0007; Appendix A) indicated significant reporting biases among the RCTs included in the NMA, whereas those for disease inflammation did not indicate any obvious reporting bias among said RCTs (*p* = 0.597; Appendix A).

## 4. Discussion

The primary goal of this study was to identify the relative efficacy of different ESWT protocols and combination treatments for pain, global function, and disease inflammation (i.e., joint inflammation) outcomes in individuals with KOA. The NMA results demonstrated that (1) medium-EFD and high-EFD ESWT alone had overall significant effects on pain reduction, global function restoration, and disease inflammation reduction relative to UC, irrespective of the specific shockwave type or follow-up duration; (2) combination treatments (i.e., ESWT plus either CPT or TCM) exhibited additional treatment efficacy in terms of pain reduction and global function compared with ESWT alone; (3) in comparison with UC, FoSW achieved greater treatment efficacy in regard to pain reduction than RaSW, particularly medium-EFD ESWT, regardless of the specific intervention regime (i.e., ESWT alone or combination treatments); (4) based on the cumulative ranking results, medium-EFD FoSW plus CPT, acupoint ESWT plus TCM, and high-EFD FoSW alone were the optimal treatment strategies for reducing pain, restoring global function, and reducing disease inflammation, respectively.

This NMA is clinically useful in the context of the vast number of available treatment strategies and compositions for middle-aged or older adults with mild to moderate KOA. The most efficacious treatment option is not able to be determined by conservative pairwise meta-analysis, especially coupling the included trials with multiarm trials [77]. By contrast, without double counting of the participants among the included RCTs, the NMA methods provide consistent estimates of the relative effects among head-to-head treatment arms along with direct and indirect comparisons [78]. Additionally, treatment efficacy ranking of all of the identified ESWT protocols which were comprised of different shockwave types (i.e., FoSW alone or RaSW alone) and EFD levels is critical for its ability to elucidate the optimal approach among different treatment options for patients with KOA. Even with the lack of head-to-head RCTs, this NMA provides clinicians with the evidence regarding the comparative effects among different monotherapies and combination treatments of ESWT for patients with KOA, especially those who have contraindicated conditions for intra-articular injections or invasive treatments.

The results of this NMA revealed that, rather than high-EFD ESWT, the medium-EFD ESWT exhibited the strongest treatment effect on pain and function outcomes, irrespective of shockwave type. Our findings are consistent with those of another study indicating that ESWT applied at a medium dosage (EFD of 0.08–0.25 mJ/mm^2^) produced greater effects in terms of pain as measured using WOMAC scores [41]. However, the metaregression results in this NMA indicated that a higher EFD predicts greater treatment effects on pain reduction and function restoration, despite statistical nonsignificance; accordingly, a high-EFD (≥2.5 mJ/mm^2^) ESWT is expected to yield greater treatment effects than medium-EFD ESWT. This inconsistency may be explained through the results indicating that a higher EFD is associated with higher ESWT treatment effects, which reach a peak at an EFD exceeding 0.32 (95% CI: 0.22–0.42) mJ/mm^2^ for pain and 0.25 (95% CI: 0.21–0.29) mJ/mm^2^ for WOMAC scores [42]. Taken together, an ESWT employed at a medium-EFD dosage of 0.8 to 0.25 mJ/mm^2^ may help achieve the optimal treatment outcome, particularly in regard to pain reduction and functional improvement.

Relative effects of FoSW and RaSW on pain reduction have been investigated and compared for musculoskeletal disorders [58,79,80,81,82] as well as KOA [41,42]. However, the results with respect to the superior treatment efficacy between the two shockwave types remain inconclusive. In this NMA, FoSW ranked the highest for pain reduction at immediate and medium-term follow-up, but RaSW had the highest ranking at short-term and long-term follow-up, particularly for the medium EFD treatment for KOA. Our findings are in agreement with the results of other studies comparing the treatment effectiveness of FoSW and RaSW subgroups in pain reduction for common soft tissue disorders of the knee [16]. We further verified that, when pooling the comparisons for all follow-up timeframes, medium-EFD FoSW plus CPT was ranked the most optimal treatment for pain reduction during the overall follow-up duration. Our finding indicates that the energy level may be the determinant of ESWT efficacy rather than the shockwave type for pain reduction.

In this study, the results of direct pairwise meta-analysis and NMA demonstrated that combination ESWT treatments incorporating CPT or TCM typically yielded greater effects for pain reduction and function restoration compared with ESWT alone, regardless of shockwave type or energy dosage. In addition, the combination treatments FoSW plus CPT and acupoint RaSW plus TCM were ranked the most effective treatment for pain and function outcomes, respectively. Our findings indicate that the combination treatment regimen of ESWT may be superior to ESWT monotherapy, which corroborates other subgroup analysis study results revealing a significant difference between adjunct treatment (SMD = −5.85) and monotherapy (SMD = −2.35) subgroups in terms of the WOMAC scores of patients with KOA [42].

Inflammatory factors are associated with KOA disease progression. Multiple studies have identified the effects of ESWT on the reduction of inflammation in animal models [83,84] and human osteoarthritic chondrocytes [85,86]. Analysis results in the present NMA revealed that high-EFD FoSW exerted significant effects on disease inflammation in terms of decrease in joint inflammation and was ranked the most optimal option among ESWT treatment arms. Our findings are consistent with those of other researchers and further indicate that ESWT, particularly FoSW, with an EFD greater or equal to 2.5 mJ/mm^2^, exerts promising effects on joint inflammation for patients with KOA.

In this NMA, a series of metaregression models was established to identify potential moderators affecting relative efficacy among treatment arms. We observed no significant moderation effects in terms of participant characteristics, methodological level (i.e., PEDro score), follow-up duration, and intervention design for all primary and secondary outcomes, with the exception of area of population and PEDro score, both of which influenced ESWT efficacy for pain reduction. Two reasons may explain such findings. First, participants of an older age may have experienced minor pain reduction after ESWT, which may explain why relatively young (mean age = 49.6 years) African patients experience greater pain reduction in response to ESWT compared with older (mean age = 61.5 years) European patients. Second, as mentioned, medium-EFD ESWT had greater effects on pain outcome compared with high-EFD ESWT, which corresponds to the association between population area and treatment effects in relation to pain. Most of the RCTs that enrolled African patients employed medium-EFD ESWT, whereas those studying European patients applied an EFD of 0.4 mJ/mm^2^; such findings further support that a medium EFD of 0.08 to 0.25 mJ/mm^2^ is the optimal intervention dosage of ESWT rather than a high EFD of ≥2.5 mJ/mm^2^.

The findings of this NMA must be interpreted considering the following limitations. First, because of the variation in the prescriptions of CPT or TCM (e.g., various exercise training programs, modality therapies, and Chinese pain medications) and ESWT application parameters (i.e., number of shocks, impulse frequency, and depth of energy), it was difficult to draw a definite conclusion regarding the effect of a specific ESWT protocol (such as dosage, number of impulses, and total number of treatment sessions) on pain reduction or function restoration. Second, all the non-ESWT comparators were pooled within a UC group. Because different non-ESWT comparators may have exerted influence in the pooled effects when all comparisons in the NMA were combined, the results must be interpreted with caution. Fourth, the estimates for treatment arms, including commination treatments of high-EFD and low-EFD FoSW, and acupoint ESWT plus CPT, were demonstrated with wide 95% CIs, which were subject to considerable uncertainty. Finally, for the treatment outcome of disease inflammation, the inadequate statistical power derived from small number of treatment arms may hinder the detection of inconsistency, despite of that inconsistency was not detected in the current NMA.

## 5. Conclusions

This NMA determined the relative efficacy of different ESWT regimens (i.e., shockwave type and energy level) and combination treatments (i.e., ESWT plus CPT or TCM) in terms of pain, global function, and disease inflammation in individuals with KOA; in addition, medium-EFD FoSW plus CPT was determined to be the optimal treatment strategy for pain reduction, whereas acupoint ESWT plus TCM and high-EFD FoSW alone were the optimal treatment options for function restoration and inflammation inhibition, respectively, regardless of the intervention type or follow-up duration. Based on the analyses results, we conclude that ESWT alone reduces joint inflammation, and a combination treatment incorporating ESWT with an adjunct treatment (such as CPT or TCM), especially medium EFD, exerts favorable effects on pain reduction and functional improvement in individuals with KOA. Moreover, the study results contribute to the knowledge of optimal ESWT intervention strategies, emphasizing the need for a combination treatment to manage pain and functional decline in individuals with KOA. The findings of this NMA provide evidence for clinicians regarding the optimal ESWT regimens to ensure successful treatment outcomes. Based on the limitations of this NMA, additional studies enrolling large number of participants are warranted to further identify specific intervention protocols.

## Figures and Tables

**Figure 1 biomedicines-10-00306-f001:**
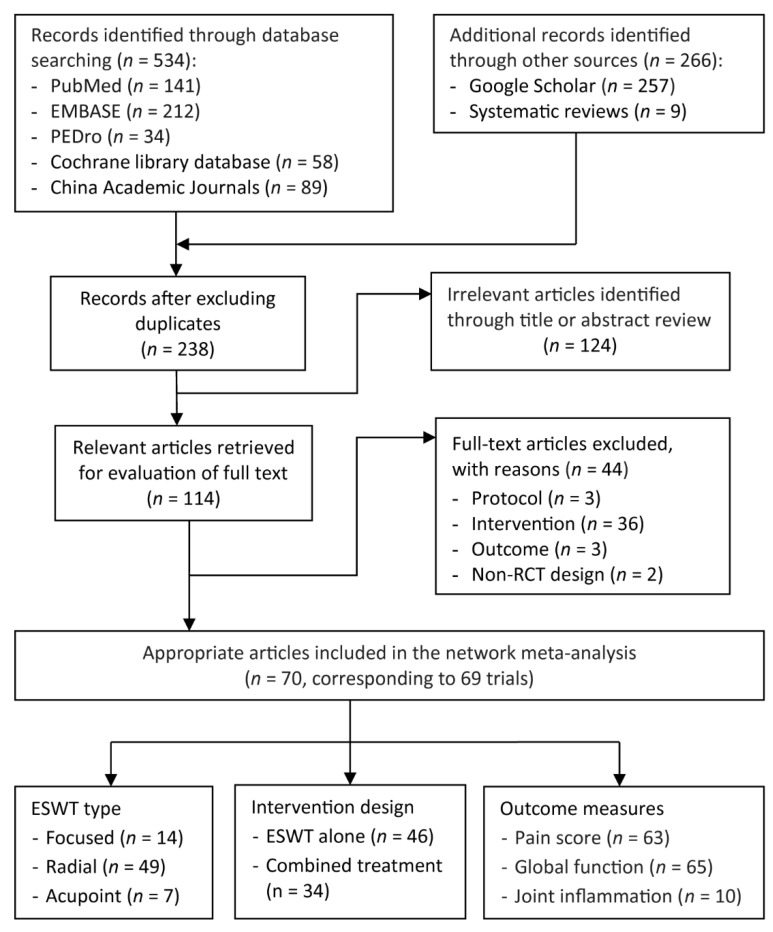
PRISMA flowchart of the study selection. ESWT, extracorporeal shockwave therapy; RCT, randomized control trial.

**Figure 2 biomedicines-10-00306-f002:**
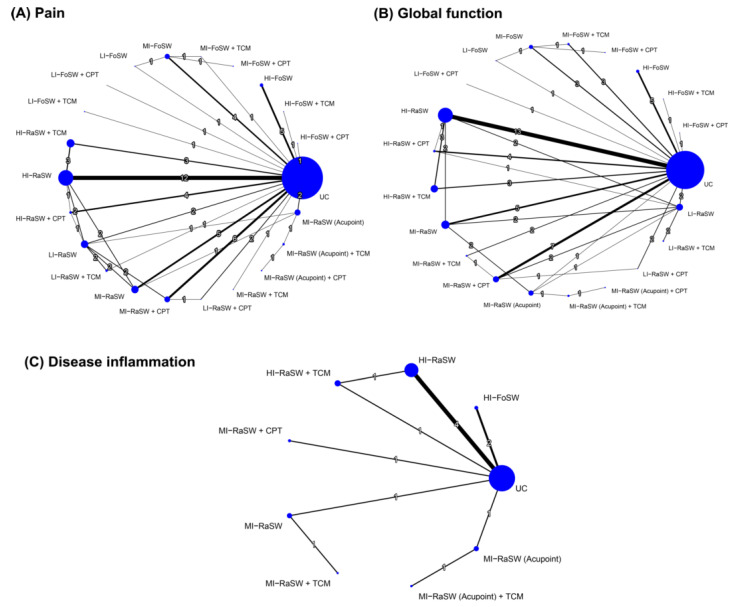
Network plot of the direct comparisons of different treatments for (**A**) pain, (**B**) global function, and (**C**) disease inflammation. The lines between nodes indicate direct comparisons in various studies. The size of each node is proportional to the number of the participants. The thickness of each line is proportional to the number of studies denoted on the line. FoSW, focused shockwave; RaSW, radial shockwave; HI, high energy; MI, medium energy; LI, low energy; CPT, conventional physical therapy; TCM, traditional Chinese medicine; UC, usual care.

**Figure 3 biomedicines-10-00306-f003:**
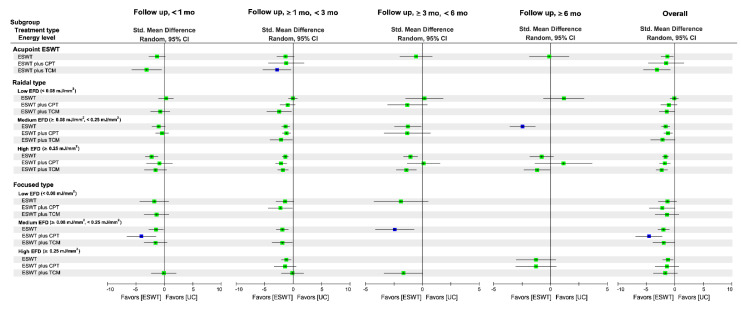
Relative effects among treatment regimens of ESWT on pain reduction at each timeframe and during an overall follow-up duration. The square point in each timeframe and for the overall duration presents the network combined effect (SMD) on changes in pain score relative to UC, and the horizontal line denotes a corresponded 95% CI. The highest rank of probability among all treatments in the same timeframe is denoted by a blue point. 95% CI = 95% confidence interval; mo, month; ESWT, extracorporeal shockwave therapy; EFD, energy flux density; CPT, conventional physical therapy; SMD, standard mean difference; TCM, traditional Chinese medicine; UC, usual care.

**Figure 4 biomedicines-10-00306-f004:**
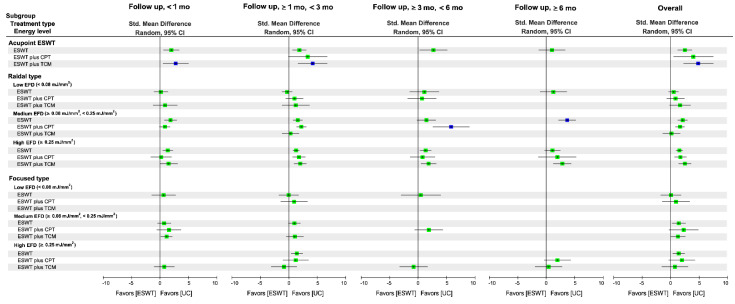
Relative effects among treatment regimens of ESWT on pain reduction at each timeframe and during an overall follow-up duration. The square point in each timeframe and for the overall duration presents the network combined effect (SMD) on changes in pain score relative to UC, and the horizontal line denotes a corresponded 95% CI. The highest rank of probability among all treatments in the same timeframe is denoted by a blue point. 95% CI = 95% confidence interval; mo, month; ESWT, extracorporeal shockwave therapy; EFD, energy flux density; CPT, conventional physical therapy; SMD, standard mean difference; TCM, traditional Chinese medicine; UC, usual care.

**Figure 5 biomedicines-10-00306-f005:**
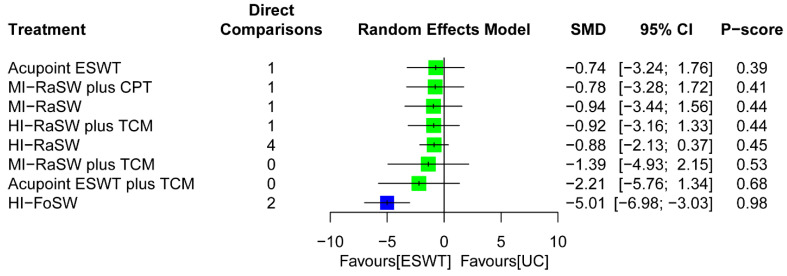
Forest plot summarizing the effects of treatment regimens of ESWT on disease inflammation for the overall follow-up duration. The blue point denotes the highest rank of probability, indicating that the treatment approach is the optimal intervention among all treatments. 95% CI = 95% confidence interval; CPT, conventional physical therapy; ESWT, extracorporeal shockwave therapy; FoSW, focused shockwave; HI, high energy; MI, medium energy; RaSW, radial shockwave; TCM, traditional Chinese medicine; UC, usual care.

**Table 1 biomedicines-10-00306-t001:** Criteria of study selection.

Trial design	Randomized controlled trial; quasirandomized controlled trial
Participant	Symptomatic or radiographic diagnosis of knee osteoarthritis
Treatment group	Received ESWT alone, ESWT plus CPT, or ESWT plus TCM
Control group	Received a placebo ESWT, relatively low-dosage ESWT, or non-ESWT intervention (i.e., CPT or TCM)
Outcome	Pain, global function, disease inflammation

CPT, conventional physical therapy; ESWT, extracorporeal shockwave therapy; TCM, traditional Chinese medicine.

**Table 2 biomedicines-10-00306-t002:** Summary of study characteristics.

	Experimental Group				Control Group				Total			
	Trials (n) ^a^	Study Arm (n)	Patients (n)	Mean (Range) ^b^	Trials (n) ^a^	Study Arm (n)	Patients (n)	Mean (Range) ^b^	Trials (n) ^a^	Study Arm (n)	Patients (n)	Mean (Range) ^b^
Age, years	68	93	3701	60.1 (40.1–80.3)	53	60	2209	59.3 (43.6–72.7)	68	153	5910	59.8 (40.1–80.3)
BMI, kg/m^2^	29	38	1275	25.8 (22.3–35.3)	23	25	784	25.7 (22.3–36.4)	29	63	2059	25.7 (22.3–36.4)
Sex, n												
Male	50	72	1242		37	38	618		57	128	2048	
Female	58	82	2113		43	45	1155		65	145	3706	
Disease duration, mo	48	69	2982	51.4 (6–200)	33	34	1443	47.6 (6–138)	53	116	4850	52.4 (6–200)
K-L grade												
≤II	26	37	1062		20	20	571		46	57	1633	
II–III	31	44	1597		24	29	974		55	73	2571	
I–III	47	66	2697		36	42	1592		83	108	4289	
≥III	3	3	98		3	4	113		6	7	211	
Involved knee, n												
Unilateral	28	36	1225		24	25	810		33	74	2280	
Bilateral	13	19	329		10	13	254		18	45	769	
Population (area)												
Europe	4	4	132		4	4	133		4	8	265	
Africa	6	7	110		5	8	130		6	15	240	
Asia	59	83	3494		45	49	1981		59	132	5475	
Intervention design (compliance, %)												
ESWT alone	46	57	2234	98.8 (88.1–100)					46	57	2234	98.8 (88.1–100)
ESWT + CPT	19	22	710	95.0 (44.4–100)					19	22	710	95.0 (44.4–100)
ESWT + TCM	15	15	792	100 (100–100)					15	15	792	100 (100–100)
Comparator type (compliance, %)												
None					4	4	187	98.0 (90.0–100)	4	4	187	98.0 (90.0–100)
Placebo					13	13	466	95.6 (83.3–100)	13	13	466	95.6 (83.3–100)
PM					11	11	546	97.8 (80.8–100)	11	11	546	97.8 (80.8–100)
CPT					21	24	652	97.4 (86.7–100)	21	24	652	97.4 (86.7–100)
TCM					9	9	393	93.0 (98.6–100)	9	9	393	93.0 (98.6–100)
Pain (10-point VAS)	49	62	2542	8.8 (4.5–8.1)	42	45	1688	6.6 (4.0–8.7)	49	107	4230	7.9 (4.0–8.7)
Global function												
WOMAC (0–100)	42	54	2217	50.4 (2.7–98.2)	34	35	1328	50.4 (2.7–98.2)	42	89	3545	49.1 (2.7–98.2)
Lequesne index (0–24)	18	22	829	11.9 (7.8–17.4)	16	16	631	11.4 (7.9–17.2)	18	38	1460	11.7 (7.8–17.4)
Lysholm index (0–100)	9	13	758	47.7 (38.1–68.1)	7	7	382	51.6 (39.7–67.5)	9	20	1140	49.1 (38.1–68.1)
Disease inflammation												
IL-1 (pg/mL)	9	11	571	75.6 (17.3–220.9)	7	7	402	49.6 (16.9–113.4)	9	18	973	65.5 (16.9–220.9)
TNF-α (pg/mL)	10	12	625	37.7 (9.1–48.3)	8	8	456	32.8 (9.1–45.3)	10	20	1081	35.7 (9.1–48.3)
Nitric oxide (μmol/mL)	5	7	489	80.3 (65.7–96.4)	4	4	280	73.1 (64.2–76.4)	5	11	769	77.7 (64.2–96.4)

^a^ The number of trials that reported the indicated item. ^b^ All summations calculated based on the values reported in the included trials and that could be estimated. BMI, body mass index; CPT, conventional physical therapy; ESWT, extracorporeal shockwave therapy; IL-1, interleukin 1; K–L grade, Kellgren and Lawrence grade; PM, pain medicine; TCM, traditional Chinese medicine; TNF-α, tumor necrosis factor α; VAS, visual analogue scale.

**Table 3 biomedicines-10-00306-t003:** Abbreviations for treatment arms.

Treatment Arm	Abbreviation
Acupoint therapy using ESWT	Acupoint ESWT
Acupoint ESWT plus CPT	Acupoint ESWT + CPT
Acupoint ESWT plus TCM	Acupoint ESWT + TCM
Radial shockwave	RaSW
High-energy radial shockwave	HI-RaSW
Medium-energy radial shockwave	MI-RaSW
Low-energy radial shockwave	LI-RaSW
High-energy radial shockwave plus CPT	HI-RaSW + CPT
Medium-energy radial shockwave plus CPT	MI-RaSW + CPT
Low-energy radial shockwave plus CPT	LI-RaSW + CPT
High-energy radial shockwave plus TCM	HI-RaSW + TCM
Medium-energy radial shockwave plus TCM	MI-RaSW + TCM
Low-energy radial shockwave plus TCM	LI-RaSW + TCM
Focused shockwave	FoSW
High-energy focused shockwave	HI-FoSW
Medium-energy focused shockwave	MI-FoSW
Low-energy focused shockwave	LI-FoSW
High-energy focused shockwave plus CPT	HI-FoSW + CPT
Medium-energy focused shockwave plus CPT	MI-FoSW + CPT
Low-energy focused shockwave plus CPT	LI-FoSW + CPT
High-energy focused shockwave plus TCM	HI-FoSW + TCM
Medium-energy focused shockwave plus TCM	MI-FoSW + TCM
Low-energy focused shockwave plus TCM	LI-FoSW + TCM

CPT, conventional physical therapy; ESWT, extracorporeal shockwave therapy; TCM, traditional Chinese medicine.

## Data Availability

Refer to Appendix A. Raw data available on request.

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
