# Peer review of "Relative Effect of Extracorporeal Shockwave Therapy Alone or in Combination with Noninjective Treatments on Pain and Physical Function in Knee Osteoarthritis: A Network Meta-Analysis of Randomized Controlled Trials"

_biomedicines, 2022, doi:10.3390/biomedicines10020306_

Round 1
Reviewer 1 Report
Dear authors,
Congratulations for this systematic review! It is very well written and highlights an important worldwide issue: the treatment of knee osteoarthritis, a disease with an increasing number of cases, due to the growth tendency of population aging. I have only a few minor recommendations for you:
- I think it will be useful to add in the introduction a short paragraph about ESWT cost effectiveness, in order to highlight the importance of this valuable technique.
- Line 126: use italic for in vitro and in vivo.
- Please be careful and use justify in all the manuscript (e.g., lines 160-173, 189-197, 202-216).
- Figure 2 – Maybe it will be a good idea to rearrange figure 2 ( A and B on a line and C centrated under them).
- Being such a complex meta-analysis, I suggest you to add an Abbreviations section.
Author Response
[Biomedicines (ISSN 2227-9059)]
Manuscript ID: biomedicines-1548718
Author's Reply to the Review Report
Comments and Suggestions for Authors
Dear authors,
Congratulations for this systematic review! It is very well written and highlights an important worldwide issue: the treatment of knee osteoarthritis, a disease with an increasing number of cases, due to the growth tendency of population aging. I have only a few minor recommendations for you:
- I think it will be useful to add in the introduction a short paragraph about ESWT cost effectiveness, in order to highlight the importance of this valuable technique.
Response
Thank you for your comprehensive review and constructive comments. Following the reviewer’s comments, we made the statement as follows:
Page 2, Lines 52–53.
“Extracorporeal shockwave therapy (ESWT) is a convenient, cost-effective treatment for managing pain in common musculoskeletal conditions of the lower limbs [14-18].”
- Line 126: use italic for in vitro and in vivo.
Response
The statement was corrected as follows:
Page 3, Lines 121–123.
“(2) the trial was conducted in vitro or in vivo by using an animal model;”
- Please be careful and use justify in all the manuscript (e.g., lines 160-173, 189-197, 202-216).
Response
The statements were revised as follows:
Page 4, Lines 155–157.
“If the RCT separately reported treatment effects on bilateral legs, those of bilateral legs were combined to enable a single comparison, and such procedure has been recommended in Cochrane Handbook for Systematic Reviews of Interventions [60].”
Page 4, Lines 173–187.
“In this NMA, the PEDro quality score was used to rank the methodological quality and risk of bias. Two of the team members, CDL and SWH, independently assessed methodological quality of each included RCT. The PEDro scale comprises 10 ranking items which …. Validity of the PEDro scale has been identified [65]. The interrater reliability of ratings for the individual PEDro scale items varies from moderate to excellent (Kappa value: 0.53-0.94) for assessing the quality of RCTs [66]. In addition, an intraclass correlation coefficient for the PEDro total sum score has been identified as 0.91 [95% confidence interval (CI): 0.84–0.95] [67]. The methodological quality of the included RCTs was considered as low, medium, and high with a total PEDro score ≤3/10, 4–6/10, and ≥7/10, respectively [68].”
- Figure 2 – Maybe it will be a good idea to rearrange figure 2 (A and B on a line and C centrated under them)
Response
Figure 2 was revised according to the reviewer’s comment.
- Being such a complex meta-analysis, I suggest you to add an Abbreviations section.
Response
We added an Abbreviations table (Table 3) in section 3.3.2.

Reviewer 2 Report
Dear authors,
your review entitled “Relative effect of extracorporeal shockwave therapy alone or in 2 combination with noninjective treatments on pain and physical 3 function in knee osteoarthritis: A network meta-analysis of 4 randomized controlled trials” identify the effects relative to the different ESWT regime and combination treatments on pain and functional outcomes in individuals with KOA. This is an interesting review, but some minor revisions are needed before its publication.
Introduction
-please explain better the purpose of the work.
Materials and Methods
-add a table to the paragraph "Selection criteria of studies", makes it more immediate to understand
Figure
-Figure 2 :improves the quality of figure 2 is not clear
Author Response
[Biomedicines (ISSN 2227-9059)]
Manuscript ID: biomedicines-1548718
Author's Reply to the Review Report
Comments and Suggestions for Authors
Dear authors,
your review entitled “Relative effect of extracorporeal shockwave therapy alone or in 2 combination with noninjective treatments on pain and physical 3 function in knee osteoarthritis: A network meta-analysis of 4 randomized controlled trials” identify the effects relative to the different ESWT regime and combination treatments on pain and functional outcomes in individuals with KOA. This is an interesting review, but some minor revisions are needed before its publication.
Introduction
-please explain better the purpose of the work.
Response
Thank you for your comprehensive review and constructive comments. Following the reviewer’s comments, we made the statement as follows:
Page 2, Lines 83–88.
“The relative effects among different combination treatment regimens of various ESWT applications remain unclear. Therefore, the purpose of this study was (1) to identify the relative effects of different ESWT applications and combination treatment regimens on pain outcome, global function, and disease inflammation through network meta-analysis (NMA) and (2) to determine the optimal treatment strategy by using the ranking probabilities of each intervention type for individuals with KOA.”
Materials and Methods
-add a table to the paragraph "Selection criteria of studies", makes it more immediate to understand
Response
We added Table 1 in section “2.3. Selection criteria of studies”.
Figure
-Figure 2 :improves the quality of figure 2 is not clear
Response
Figure 2 was revised.

Reviewer 3 Report
The authors have conducted meta-analysis on the effects of ESWT on pain, physical function , and inflammation in patients with knee OA.
Comments
- All the typos should be corrected.
- Lines 43-45: This sentence is not clear and incorrect. This should be corrected.
- Line 145: It is not clear why the authors have chosen chemerin for KOA disease activity assessment and did not include other adipokines. This should be explained.
- Lines 141: The authors should define exactly the difference between “the KOA disease activity” and “disease inflammation”, and “inflammation” , and unify term usage in the manuscript. It should be also explained why pain is not included into KOA disease activity.
- 1 description should be presented in the Methods section. This should be corrected.
- Table 1: The patients’ characteristics should be extended: the data related to pain, physical function, and inflammation should be presented.
- Lines 421-413, 430-432, 454-456: this data should be included into corresponding Figure legend as a Note. This should be corrected.
Author Response
[Biomedicines (ISSN 2227-9059)]
Manuscript ID: biomedicines-1548718
Author's Reply to the Review Report
Comments and Suggestions for Authors
The authors have conducted meta-analysis on the effects of ESWT on pain, physical function , and inflammation in patients with knee OA.
Comments
- All the typos should be corrected.
Response
Thank you for your comprehensive review and constructive comments. We corrected the typos.
- Lines 43-45: This sentence is not clear and incorrect. This should be corrected.
Response
Page 1, Lines 43–44.
“With disease progression, KOA impairs musculoskeletal system [4], ultimately leading to physical difficulty [2, 5, 6].”
- Line 145: It is not clear why the authors have chosen chemerin for KOA disease activity assessment and did not include other adipokines. This should be explained.
Response
We revised the statement as follows:
Page 4, Lines 142–144.
“… and (3) synovial fluid adipokines such as chemerin, which are associated with the disease severity of KOA [58, 59].”
- Lines 141: The authors should define exactly the difference between “the KOA disease activity” and “disease inflammation”, and “inflammation” , and unify term usage in the manuscript. It should be also explained why pain is not included into KOA disease activity.
Response
We revised to use the term “disease inflammation” throughout manuscript.
- 1 description should be presented in the Methods section. This should be corrected.
Response
Table 2 was inserted in the Methods section (below the section 2.5).
- Table 1: The patients’ characteristics should be extended: the data related to pain, physical function, and inflammation should be presented.
Response
We added the data related to pain, global function, and disease inflammation in Table 2 (original Table 1).
- Lines 421-413, 430-432, 454-456: this data should be included into corresponding Figure legend as a Note. This should be corrected.
Response
Figure legends of Figure 3-5 were corrected.
